# A Series of Happenstances: How the Pandemic Created Opportunities with Technology

Jean Kiekel [1], Jennifer Courduff [2,*] and Peter Hessling [3]

1   School of Education, University of Saint Thomas, SEHS, 3800 Montrose, Houston, TX 77006, USA
2   School of Education, Azusa Pacific University, 901 E Alosta, Azusa, CA 91702, USA
3   College of Education, North Carolina State University, 2211 Hillsborough St., Raleigh, NC 27607, USA
*   Correspondence: jcourduff@apu.edu

**Abstract:** The Happenstance Learning Theory (HLT) has primarily been used in career counseling fields, predominantly in helping clients transition through different happenstances in life to find a career path. However, the COVID-19 pandemic showed that our nation's teachers also developed the skills that are considered the pillars of HLT, as they made the transition from teaching traditionally one day to remote learning the next. This study explored how 46 special educators adapted their teaching methods to best address the needs of their students despite challenging circumstances. This study also suggests the application of the Happenstance Learning Theory in understanding and explaining how teachers seemingly make changes "in the moment".

**Keywords:** persistence; special education; happenstance learning theory





## 1. A Series of Happenstances: How the Pandemic Created Opportunities with Technology

The COVID-19 global pandemic created a unique time in education. Educators around the world were immediately required to transition from traditional teaching measures to teaching online. What they learned during the experience has spawned much research. Most of this research is based on the tools that were used to teach traditional K-12 students, with little being known about the subspecialties of education, such as special education. This paper applies a human resources theory to the experiences of a group of special educators and examines how they coped and dealt with the pivot to emergency remote teaching.

## 2. Literature Review

The novel coronavirus, or COVID-19, global pandemic affected educators worldwide. In the United States, that effect began to be felt in March 2020. Because of its highly contagious nature, health organizations recommended closing schools and other places where large gatherings occur in an attempt to reduce the possibility of transmission. Even with these measures, approximately 80 million people contracted the virus, with nearly 100,000 deaths in the United States [1]. The closure of schools affected more than 90% of the student population [2]. The measures taken to prevent the spread of the virus were devastating to education, especially at the K-12 level. While the effects of the pandemic were felt throughout K-12 and higher education, some populations of students were affected more so than others, especially those often-overlooked populations of students and teachers. This led this research team to focus on the effects felt by special educators, one such underrepresented group of educators.

Educators at the K-12 level were highly affected by the resulting pivot to emergency remote teaching. This was primarily because this group of educators are not trained in online pedagogies as a regular part of their educator preparation [2,3]. The ability to teach remotely requires the development of a different set of pedagogies than teaching face-to-face [4]. Even the type of distance learning delivered during the pandemic was very

different from traditional online learning. What happened during the pandemic has been called emergency remote teaching (ERT). Some of the major differences between emergency remote teaching (ERT) and traditional online teaching are as follows: ERT is designed to be temporary and is caused by a state of emergency to replace teaching that would usually be carried out in a more traditional, in-person setting. It is also understood that, as soon as things return to normal, instruction will return to its more traditional format [5]. ERT is not optional, whereas traditional online teaching often is. ERT is not well-planned in advance of its implementation, including the training of teachers and students to be successful in an online environment. Those who are trained in online pedagogies and have been successful in online teaching understand what it takes to facilitate learning via an online environment [4,6,7]. These differences between ERT and traditional online teaching are like night and day. For the most part, teachers attempted to copy their face-to-face (F2F) teaching methods in the online classroom with varying degrees of success [4]. All students in K-12 struggled with the sudden shift to online learning, but students with special needs were especially hard hit [8].

Students with special needs rely on their caregivers for the support and structure they need. These caregivers include parents, teachers, and therapists (Scheffers et al., 2021). Teaching students with special needs requires structure, the maintaining of routines, and support, which were disrupted during the pandemic. This caused educators to scramble to find ways to re-establish these protocols for their students [2]. What caused even more difficulty was that students with special needs often require the services of more than one caregiver. In addition to a teacher, there may also be a need for occupational and/or physical therapists, speech/language pathologists, and mental health professionals [2]. In addition, laws regarding students with special needs require that Individualized Education Plans (IEPs) must be continued, even when students are out of school for extended periods of time [9].

*Happenstance Learning Theory (HLT)*

HLT states that "human behavior is the product of countless numbers of learning experiences made available by both planned and unplanned situations in which individuals find themselves. The learning outcomes include skills, interests, knowledge, beliefs, preferences, sensitivities, emotions, and future actions" ([10], p. 135). Happenstance learning can be described in two ways: instrumental learning, which occurs when we observe the consequences of our actions, positive or negative, and associative learning, which happens when we connect what is happening in the environment and the behavior we are observing as being associated. These circumstances play a major role in how we react to the happenstance, and they influence what we do to regain some semblance of control [10].

Educators, in the wake of the pandemic, found themselves in circumstances over which they had no control, requiring individuals to try to find opportunities arising during the pandemic and to find ways to capitalize on those circumstances in order to meet the needs of their students. The hallmark skills of HLT that facilitate both instrumental and associative learning are persistence, flexibility, optimism, and risk-taking [11]). Persistence is the ability to keep trying, even when experiencing setbacks. Flexibility requires the changing of attitudes and circumstances. Optimism means being able to find hidden opportunities and to see them as possibilities. Risk-taking is the courage to take action, even when not knowing what the outcome will be. The authors found that HLT research had only been published with regard to career counseling and job transitions. Our study hopes to expand the application of this theory by applying it to how teachers adapted their teaching to the pandemic.

## 3. Research Design

This research was undertaken to examine the lived experiences of PK-12 special educators and their experiences during the pivot to fully online instruction. The researchers used a phenomenological approach in order to understand the challenges faced by this

group of educators [12]. The findings from this study can be used by educator preparation programs to understand the hardships of this group of educators and to find ways to ensure that they are better prepared to use technology and other distance means for the purpose of serving the special needs student population.

**Research Questions:** What were the challenges of special educators during the pandemic? How did the circumstances arising from the pandemic demonstrate persistence, creativity, and passion in meeting those challenges?

**Research Methodology.**

This study employed a phenomenological qualitative methodology designed to understand the experiences of K-12 special educators in the United States during the pivot from traditional instruction to virtual instruction [12,13]. The pandemic brought many changes to K-12, but the reason for concentrating on this group of educators was because their challenges were unique due to their student population. As the researchers coded the interviews, the findings illustrated the characteristics of Happenstance Learning Theory, leading us to explore this interpretive lens to understand how this group of educators acted under the duress of the pandemic.

*3.1. Instrumentation, Data Collection, and Analyses*

The data for this study were collected using personal, recorded interviews and focus groups as valid data sources [14]. All interviews were carried out by one member of the research team to ensure consistency in the interview process. These interviews were completed over a six-week period in late 2020. All participants were asked the same questions in the same order as part of the interview protocol. There were 46 participants in the study. All participants were somehow affiliated with special education, either as a special education teacher or as occupational or physical therapists, speech/language pathologists, and resource specialists. Two focus groups were held, with all members of the research team participating in order to triangulate the data collected during the interview process. The interviews were conducted via Zoom and transcribed. The questions asked during the interviews were meant to discern the difficulties and successes of the participants in working with students with special needs. These included questions related to demographics; experiences before, during, and after the pandemic; what the educators learned about themselves and their abilities; and their take-aways from the experiences that they will continue as we come away from the experience of the pandemic. The focus group questions centered around validating and clarifying the themes that the research team identified.

While Seidman [15] and others recommend multiple interviews for each participant when conducting phenomenological research, this was not possible given the number of individual interviews that were conducted, the logistics of the interviews (conducted on Zoom by one researcher), and the national scope of the study. Efforts were made in each interview to address context, the details of the educators' experiences, and their reflections on the meanings of their experiences. While we were unable to conduct multiple interviews with each participant, we believe that we were still able to capture how participants in multiple locations and roles perceived their experiences in trying to teach during a pandemic.

The research team individually coded the interviews and held twice-monthly meetings to discuss the codes and cross-codes of significant statements. We then combined the codes into themes [12]. The themes were then developed into textural and structural descriptions to help the research team understand the essence of the experiences. In order to maintain methodological and analytic rigor, the researchers consistently shared field notes and analytic memos. The researchers also used member checking so that individual participants could verify and clarify their interview transcripts. Member checking [16] is a qualitative method of enhancing the validity of a study. We believe that our approach to this study—and the sheet number and diversity of the interviews—helped lead to an accurate, holistic, and macro-view of the participants' experiences.

*3.2. Participants*

Calls for participation were sent out through the International Society of Technology in Education (ISTE) Commons, the ISTE Teacher Educator Network (ISTETEN), ISTE Twitter, and the Quality Indicators for Assistive Technology (QIAT) listserv. There were forty-six special educators who answered the call. The respondents were located nationwide and included teachers who teach students with cognitive, behavior, social/emotional, and physical disabilities; speech/language pathologists; occupational therapists; assistive technology specialists; physical therapists; and applied behavior analysis therapists. Pseudonyms were created for anonymity of reporting. See Appendix A for participant demographics.

## 4. Findings

This research was undertaken to discover how an underrepresented group of educators were affected by the pandemic. Special educators are a unique group of educators whose student population requires more hands-on, direct teaching methods to facilitate their learning. As the research team began to code and cross-code the data, many themes emerged. The major themes included technology, uncertainty, professional development/preparation, and compliance. In our twice-monthly discussions, we discovered that, oftentimes, our participants seemed to have developed skills that were related to HLT. The latter finding led us to look more closely at the elements of HLT and how they related to what we found.

*4.1. Technology*

Technology was a major theme for nearly all participants. Much of what occurs in a special needs' classroom involves hands-on activities performed with participation from the students. However, the online classroom meant that hands-on activities were not possible. Marcia said, "We didn't even have a projector in the classroom to use [pre-pandemic], so I wasn't really using technology that much in my instruction". Ruth stated, "Technology is not my forte. The thought about teaching virtually terrified me". Jill added, "Many teachers are technophobic, others not so technophobic, but they weren't ready for this".

The types of technology available to the participants and their students also played a role in the teachers' abilities to work remotely with their students. Norah noted, "Chromebooks do not allow for the screen sharing feature and the operating system we use works on Chromebooks. For my students who have their own personal devices or an iPad, I can bring something up on the screen, I can give them the mouse, and they can write on my screen. On any district Chromebook, I cannot do that".

The digital divide also created major problems with using technology for these educators. Kimberly noted, "You might have four kids in a family and they're all sharing a cell phone to do to their work and prioritizing—oh, this child is in special ed, he can just not get any right now. Johnny's a junior in high school and gets priority over you just sitting and listening to a story". Jill added, "A lot of the kids did not have their own devices at all. A lot of them didn't have internet connectivity". She went on to state that there were kids in the district who not only did not have devices or internet connectivity but also did not speak English as their primary language, which made communication sometimes difficult.

*4.2. Uncertainty*

All participants expressed the uncertainty caused by the pandemic. Diane stated that she was told, "Keep it together as best you can. You're going to try and make a bunch of asynchronous type packets that can go home. At that time in special education there was no, 'you must be doing asynchronous learning' just because we couldn't necessarily figure out how to do it". Stephanie noted, "I wasn't at all prepared to give kids anything and we had 10 min because school's out at 2:20 and we heard this [school was closing] at 2:00". Even as the spring semester ended, the uncertainty continued throughout the summer and even into the fall. Amy stated, "A lot of energy was spent on the what ifs—how do I control this or monitor our community, my family, my extended family, and in some ways it's almost like, well, I can't really think too far ahead. I felt, and still feel like, things are

still changing and that to sit and plan all summer for the who school year was no longer a relaxation event".

### 4.3. Professional Development/Preparation

As previously stated, K-12 educators are not necessarily prepared for virtual teaching and learning, and this was a subtheme of technology. While districts worked to provide professional development for their teachers, Norah noted, "Often they're [trainings] not SPED-specific". Because things were changing so rapidly with students, teachers, school policies, etc., many teachers sought their own training, as Amy stated:

"How I was trained was my own doing. I think the district tried to provide some things, but really, it was, I had to go out and find what I needed to because they [district trainers] would say, 'oh, this is what elementary school teachers need' . . . or we'd watch something for the high school and they said, 'well, you don't need to see this', and I'd sneak in because it was an open link. And they said 'you can choose, but you probably don't need it' and I watched it, and said 'wait a second, that fits exactly with what I need."

Stephanie reinforced this idea of the lack of formal training and preparation, saying, "I can't figure it out so I'm going to look on YouTube. How do I do this? So I look at Facebook, I look at YouTube".

Participants also commented about the need for student training. The term digital natives has been used to describe today's students, but Kimberly made a different observation about today's students:

"Students are digital tourists . . . Have you ever been on a tour, like during vacation, where you had a tour guide? Did you have to think during that? The tour guide tells you 'go left', ok, go left. They tell you a little thing, then you go right. Our students are digital tourists. They know the latest, greatest, and coolest apps out there . . . they can do the basics of the tools, but they don't know how to use the tools."

Technology is constantly changing. As the participants became more familiar with technology and it began to feel more effective, new technology tools were developed that seemed to show more promise. Amy noted, "It was evident as you were watching [training videos] that those are really good strategies, and then you realize how quickly our education setting or our therapy settings just evolved so quickly that some of the information that I was reading or learning about in June was already outdated".

### 4.4. Compliance

Compliance issues are important in the special education context. All students with special needs have individualized education plans (IEPs), which detail the type of instruction, support, and services that must be provided to students with special needs in order for them to be successful in the educational setting. The US Department of Education reinforced that education services were to be continued during the pandemic with this statement: "If a student who has an individualized education program (IEP) through the Individuals with Disabilities Education Act, or is receiving services under Section 504, is required or advised to stay home by public health authorities or school officials for an extended period of time because of COVID-19, provision should be made to maintain education services." [9], p. 23. This edict made it clear that this group of educators had to find a way to continue to provide services at a distance. Providing those services became a major concern for the participants. Roberta stated, "We must provide learning experiences and engagement opportunities, but the shape of what that would take was very much loosely defined . . . Provision of therapy was not something our state was requiring. We were just charged with making connections and seeing our students in some form or capacity". Educators, such as Amy, also questioned whether they could legally have two students in the same Zoom session, stating "I didn't know how I could ethically put two

kids in the same [Zoom] room and have parents from other families witnessing what was on the screen". Therefore, participants carried out a lot of one-on-one sessions throughout their day.

The use of ERT gave educators a glimpse of their student's home lives, which they had previously been unaware of. Kimberly stated, "Virtual learning is giving us a glimpse into the home that we didn't know ... their home life that they were trying to hide". Some of the participants noted that they developed a stronger relationship with their parents because of the ERT. Norah stated, "I definitely have had parents really, I have had them show appreciation for my ability to understand where they're at and they don't have it all together and that's ok". Many teachers expressed empathy with what the parents and families were going through at this time. Marcia said, "I really felt for the families, but I would say, by the end of it, we had a really good system in place and the families were very supportive".

### 4.5. HLT-Related Themes

As the educators worked through the various issues that arose during their pivot to remote learning, they discovered a great many things about themselves and their ability to teach their student population that showed the development of the HLT skills outlined by Kim et al. (2016) of persistence, flexibility, risk-taking, and optimism.

### 4.6. Persistence

Pandemic teaching was difficult on all accounts. Educators at the K-12 level, in general, struggled due to a lack of training and preparation before being thrust into ERT. Special educators were even more disadvantaged because most did not use technology of any sort in providing services to their students. Many students with special needs require one-on-one support and/or support from multiple providers [2,8,17]. However, they kept at it, even when they did not know how it would work. Marcia explained, "Like, you know, this is what I believe we're doing, and this is how it's all going to work, when I didn't really know what we were doing. It's like I'm going to pretend like I know exactly what's going on, but I really don't know". Kimberly, who was an independent contractor for several districts, stated,

> "I contacted every single one of my districts and I say, look, this company has now released all of us from our responsibilities (that was their exact words). I said, I am not leaving your teachers and your families high and dry. I'll work. You don't have to put me on your payroll; you don't have to pay me, I said. But your teachers are going to need someone that they can talk to about assistive technology."

These teachers continued to persist in finding things that would work for their students, as well as for their colleagues.

### 4.7. Flexibility

The concern for student privacy and ethical decisions often required the participants to be very flexible with their schedules. They worked early in the mornings to late in the evenings in order to accommodate their students. Norah shared how she worked with one student: "I have a session scheduled for five o'clock at night because his parents really want it [services], but they work all day. I'm willing. I technically don't have to do that because it's not in my contract. I have flexibility in my contract. I work eight hours a day but at the same time, it's a little crazy". Some teachers found that they could be more flexible and that everything would be ok. Ruth stated, "I've learned to be more flexible, and I can be very type A".

### 4.8. Risk-Taking

Being unprepared for the abrupt shift and the fact that participants were not regular users of technology meant that they were suddenly forced to learn and integrate technology

into their teaching regimen. Diane expressed the reasons for the difficulty: "With life skills, so much of everything is hands-on. How do you do hand-over-hand when you can't touch the kid to help them out". This required the participants to try things not knowing how, or if, they would work. Martha shared, "It's been a lot of trial and error". They turned to a number of resources to find things. Erica's risk-taking led her to a number of tools that she felt really would improve her ability to teach and her student's abilities. She stated, "They are going to go so much further now that they have these tools too". The risk-taking of the participants benefited them and their students in ways that they had not thought possible. As Alexis discovered, "This is the power of technology, your ability to give choice. Give a variety of ways to show what you know".

*4.9. Optimism*

Ultimately, even though the participants shared that they experienced uncertainty, chaos, lack of support, etc., they discovered things about themselves that led them to be optimistic. Ruth stated, "I felt like I was being more effective and actually teaching rather than just kind of herding sheep". She also added that "I feel I am better for it, but it certainly has not been fun to go through". Norah became inspired, saying, "I'm inspired to try to partner with parents more on some of these things". Kimberly stated, "I thought COVID was going to be the worst thing for my child, but it has proven to be the best thing ever".

Some of the participants also found new relationships with colleagues and felt more valued by their schools. Alexis stated, "So we all kind of said, hey, we're not getting what we need. Let's explain to you what we need. And we're going to put pressure on you to make that happen. So that's actually been awesome". Jill stated, "I'm not happy about the circumstances for this, but I feel like finally I'm being appreciated, and my talents are being used, and I'm happy". Finally, Martha stated, "My school treats us, like they call us the heartbeat of the school".

## 5. Discussion

The pandemic created a state of chaos for K-12 educators, especially for educators who were not accustomed to using technology in their teaching. Much of the available research confirms the fact that the shift to ERT left many educators in the lurch [2,4,5,18–21], as they struggled to learn to use technology tools that they were not familiar with and to adapt them for use with their student populations.

The speed of the transition was so fast that school districts were unable to put the proper support in place. The advancements in technology over the years proved to be useful during the pandemic, but the issues associated with school closures and the complete reliance on technology proved to be especially challenging [21]. Most teachers were told that the switch was going to occur with less than 24 hours to prepare; many vividly remember the date—Friday, 13 March 2020—as readily as remembering other famous dates in history.

While online learning is not new to education, for a large number of K-12 teachers, it was. ERT changed their role as a teacher [22]. The lack of preparation for online learning should be of concern to educator preparation programs nationwide, not only in preparation but also in offering professional development [23]. The lack of including sufficient technology means that teachers often feel underprepared to effectively integrate technology into their teaching.

As demonstrated by our participants, the circumstances created by the pandemic resulted in opportunities for these educators to learn something about themselves and their abilities. The opportunities that arose required them to take action and to evaluate the outcomes of such action, which resulted in persistence. Every action undertook involved some level of risk-taking. Exploring different actions to generate potential solutions required flexibility. Finally, when these educators saw successes, they were encouraged to keep going, resulting in optimism. These skills made these educators believe that serving

students with special needs online was possible, and they began taking steps to keep the successes going [24].

The participants in this study were recruited from ISTE communication networks and the QIAT listserv which is a limitation of the study. The membership requirement of these organizations limited the participants to members. The participants were from a nationwide sampling, which the research team felt was enough for the findings to be generalizable, and data sources were seen as valid sources capable of providing reliable data.

## 6. Conclusions

The Happenstance Learning Theory, at least in part, explains how the actions taken by the educators, mostly through their own doing, completely changed the way they worked with their students. HLT recognizes that people are challenged in their response to crisis situations. It recognizes that there are opportunities hidden in unplanned circumstances that can be exploited, and our educators found these opportunities, built on their small successes, and grew in their personal abilities [24].

While this pandemic is not something that has been seen in 100 years, there is the consensus that it will not be the last. Educators may have to transition to online teaching in the future [25,26] whether it be long term, such as during a pandemic, or short term, such as during a weather-related disaster. The K-12 segment is the fastest growing subdivision of education that is moving to online learning [27]. It is therefore important to address the failings that were brought about by the pandemic and to prepare future educators to be able to make the transition when called upon [28]. Because technology efficacy is influenced by educator preparation, it is up to teacher educators to become proficient enough to model effective technology use, online pedagogy, instructional design, and experience in order for pre-service teachers to effectively meet the challenges of teaching and learning in online environments [27].

## 7. Future Research

The pandemic has opened a floodgate in terms of potential future research opportunities. The effects of ERT have yet to be discovered [29]. This area of research should be undertaken as a way to provide guidance to educator preparation programs as to what is needed to ensure that K-12 teachers can effectively pivot to online learning when it becomes necessary. The research team has also discussed following up with the participants who took part in our research as things return to normal to see if they continued to use the tools that they discovered or if they reverted to their prior methods of teaching. Another area of research that was discovered during our analysis of the research but that was not discussed here is related to teacher well-being and support. Many of our participants questioned their worth, and some left the field. Studying well-being and support issues may help districts identify ways to keep teachers who became overwhelmed and discouraged during the pandemic, at a great loss to the field.

The application of HLT to education was seen as a novel application because the literature primarily uses it in career counseling research. Seeing the associated HLT skills emerge in our participants made us wonder if HLT could be applied in other crisis situations. As teacher educators who prepare future teachers, research on how to introduce and ensure teachers are able to effectively transition to online learning when the need arises should also be undertaken. While there are many more areas that can be explored related to a more effective redesign of educator preparation programs, these are only a few that our team discussed.

**Author Contributions:** Conceptualization, J.C.; methodology, J.C.; validation, J.K. and P.H.; formal analysis, J.C., J.K. and P.H.; investigator, J.C.; data curation, J.C.; writing—original draft preparation, J.K.; writing—review and and editing, J.C. and P.H. All authors have read and agreed to the published version of the manuscript.

**Funding:** This research received no external funding.

**Institutional Review Board Statement:** The study was conducted in accordance with the Declaration of Helsinki, and approved by the Institutional Review Board (or Ethics Committee) of Azusa Pacific University (approval number 20-408 received 17 October 2020).

**Informed Consent Statement:** Informed consent was obtained from all participants prior to the start of the interview.

**Conflicts of Interest:** There was no conflicts of interest from any member of our research team with regards to this research.

## Appendix A

**Table A1.** Common Acronyms in Special Education.

| Acronym | Term | Definition |
|---|---|---|
| AAC | Augmentative and alternative communication | Tools that support a person who has difficulty communicating using speech (www.asha.org, accessed on 23 March 2022). |
| ABA | Applied behavior and analysis | A therapy based on the science of learning and behavior (www.autismspeaks.org, accessed on 23 March 2022). |
| AT | Assistive technology | Any item, piece of equipment, software program, or product system that is used to increase, maintain, or improve the functional capabilities of persons with disabilities (www.atia.org, accessed on 23 March 2022). |
| GenEd | General educators | Educators who teach students in multiple subject or single subject classrooms (www.ed.gov, accessed on 23 March 2022). |
| IEP | Individualized education program | Programs developed for students with cognitive, behavioral, physical, and communication disorders (www.understood.org, accessed on 23 March 2022). |
| Mild/Mod | Mild to moderate disabilities | Students have high incident disabilities (e.g. eligibility categories of autism, learning disability, emotional/behavioral disorders, language delays). Students are typically on diploma track and will be served in general education (inclusive settings). Services may use the titles of resource specialist or teachers in a special day class. Students may have variable academic performance, attending (distractible) behaviors, and/or social behavioral needs (http://www.fresnostate.edu/catalog/subjects/lit-early-biling-specl-ed/prlm-mld-m.html, accessed on 23 March 2022). |
| Mod/Severe | Moderate to severe disabilities | Students have lower incidence disabilities (e.g. eligibility categories of autism, learning disability, emotional/behavioral disorders, language delays). Students are served in a range of settings, such as center-based sites, special day classes, and some inclusive and/or integrated settings. Students may have academic, functional, communication, and vocational learning needs (http://www.fresnostate.edu/catalog/subjects/lit-early-biling-specl-ed/prlm-mld-m.html, accessed on 23 March 2022). |
| OT | Occupational therapist | Occupational therapists treat patients who have injuries, illnesses, or disabilities through the therapeutic use of everyday activities (https://www.bls.gov/ooh/healthcare/occupational-therapists.htm, accessed on 23 March 2022). |
| PT | Physical therapist | Physical therapists help injured or ill people improve movement and manage pain (https://www.bls.gov/ooh/healthcare/physical-therapists.htm, accessed on 23 March 2022). |

**Table A1.** *Cont.*

| Acronym | Term | Definition |
|---------|------|------------|
| RSP | Resource specialist program teacher | Also known as specialized academic instructor (SAI). They typically serve students with mild to moderate disabilities in a pull-out or push-in format. |
| SLP | Speech language pathologist | Educators who work with persons with articulation, fluency, expressive language, reception language, and swallowing disorders (www.asha.org, accessed on 23 March 2022). |
| SpEd | Special educators | Educators who teach students with a range of disabilities including, but not limited to, mild/moderate or moderate/severe disabilities (www.naset.org, accessed on 23 March 2022). |
| TK | Transitional kindergarten | Transitional kindergarten (TK) is a school grade that serves as a bridge between preschool and kindergarten, to provide students with time to develop fundamental skills needed for success in school in a setting that is appropriate to the student's age and development (https://www.first5california.com/en-us/articles/what-is-tk-and-kindergarten-preschooler, accessed on 23 March 2022). |
| UDL | Universal design for learning | A framework to improve and optimize teaching and learning based on scientific insights into how people learn. (www.cast.org, accessed on 23 March 2022). |

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
