# Peer review of "A Series of Happenstances: How the Pandemic Created Opportunities with Technology"

_education, doi:10.3390/educsci12110835_

Round 1

Reviewer 1 Report

Recommendations for revision:

- The theory could be derived with more research and studies. Especially the very important chapter HLT. 

- Appendix A (participant demographics) is missing from the articel. 

- Detailed information on the method and implementation, e.g. when was the study conducted (year)? 

 - More precise statistical data (e.g. k kappa) on intercoder reliability (cross-code) should be added. 

- More detailed information on methodological procedures, like anonymization, and interview structure (e.g. guidelines) should be added. 

-  Results remain descriptive. A stronger integration into current research discourses would be desirable.

Author Response

Please see uploaded document.

Jean

Reviewer 2 Report

Excellent focus on special education/educators to create a doable boundary for the study. Clearly grounded in HLT as the theoretical framework. Emergent themes were clearly identified. The associated explanation of each theme was a skillful blend of summary narrative along with key illustrative quotes. Specific recommendations for related future research were offered. I would add how many interview sessions were conducted with each participant (typically a phenomenology requires more than one interview session) and the approximate duration of each interview session. I also saw 'passion' in the overall study purpose, but I didn't see it as an emergent theme in the reporting of findings. If it didn't emerge, I would say so, in order to maintain tight alignment with the key concepts contained in the study purpose. Superbly written and highly informative about this timely topic!

Author Response

Please see uploaded document.
